Different analysis strategies of 16S rRNA gene data from rodent studies generate contrasting views of gut bacterial communities associated with diet, health and obesity

Garcia-Mazcorro Jose F. josegarcia_mex@hotmail.com 1
Kawas Jorge R. 2
Licona Cassani Cuauhtemoc 3
Mertens-Talcott Susanne 4
Noratto Giuliana 4
1 Research and Development, MNA de Mexico , San Nicolas de los Garza , Nuevo Leon , Mexico
2 Faculty of Agronomy, Universidad Autónoma de Nuevo León , General Escobedo , Nuevo Leon , Mexico
3 School of Engineering and Sciences, Tecnologico de Monterrey , Monterrey , Nuevo Leon , Mexico
4 Department of Food Science and Technology, Texas A&M University , College Station , TX , United States of America
Gelfand Mikhail
Electronic publication date: 2020 Nov 17
Publication date: 2020
Volume: 8
Electronic Location ID: e10372
Received 2020 Jun 23; Accepted 2020 Oct 26
Copyright: ©2020 Garcia-Mazcorro et al.
Copyright year: 2020
Copyright holder: Garcia-Mazcorro et al.
License: This is an open access article distributed under the terms of the Creative Commons Attribution License, which permits unrestricted use, distribution, reproduction and adaptation in any medium and for any purpose provided that it is properly attributed. For attribution, the original author(s), title, publication source (PeerJ) and either DOI or URL of the article must be cited.
License URL: https://creativecommons.org/licenses/by/4.0/

Keywords: Gut microbiota, 16S rRNA gene, Diet, Obesity, Health

Funding: The authors received no funding for this work.

==============================
Background

One of the main functions of diet is to nurture the gut microbiota and this relationship affects the health of the host. However, different analysis strategies can generate different views on the relative abundance of each microbial taxon, which can affect our conclusions about the significance of diet to gut health in lean and obese subjects. Here we explored the impact of using different analysis strategies to study the gut microbiota in a context of diet, health and obesity.

Methods

Over 15 million 16S rRNA gene sequences from published studies involving dietary interventions in obese laboratory rodents were analyzed. Three strategies were used to assign the 16S sequences to Operational Taxonomic Units (OTUs) based on the GreenGenes reference OTU sequence files clustered at 97% and 99% similarity.

Results

Different strategies to select OTUs influenced the relative abundance of all bacterial taxa, but the magnitude of this phenomenon showed a strong study effect. Different taxa showed up to 20% difference in relative abundance within the same study, depending on the analysis strategy. Very few OTUs were shared among the samples. ANOSIM test on unweighted UniFrac distances showed that study, sequencing technique, animal model, and dietary treatment (in that order) were the most important factors explaining the differences in bacterial communities. Except for obesity status, the contribution of diet and other factors to explain the variability in bacterial communities was lower when using weighted UniFrac distances. Predicted functional profile and high-level phenotypes of the microbiota showed that each study was associated with unique features and patterns.

Conclusions

The results confirm previous findings showing a strong study effect on gut microbial composition and raise concerns about the impact of analysis strategies on the membership and composition of the gut microbiota. This study may be helpful to guide future research aiming to investigate the relationship between diet, health, and the gut microbiota.

Introduction

The digestive tract of humans and other animals is inhabited by trillions of microbes and viruses that have evolved with their host as a unit throughout millennia. Host genetics is one of the most important factors shaping the gut microbiota (Bonder et al., 2016; Dąbrowska & Witkiewicz, 2016; Zhao, Irwin & Dong, 2016; Knowles, Eccles & Baltrunaite, 2019; Suzuki et al., 2019) but environmental factors may dominate over genetics in some circumstances (Muegge et al., 2011; Rothschild et al., 2018). Among all environmental factors that can modulate the gut microbiota, diet and dietary patterns have the strongest potential to do so. For example, diet can output similar microbiota functions across mammalian phylogeny (Muegge et al., 2011) and certain diets can induce and perpetuate obesity, a phenomenon that is closely interlinked with the gut microbiota (Backhed et al., 2004; Turnbaugh et al., 2008).

Several food ingredients such as polysaccharides and polyphenols have been reported to influence lipid metabolism by altering gut microbiota composition. However, there are still many unknowns about the effect of diet and other factors on the gut microbial ecosystem and host’ health. For instance, microbes display high levels of cell-to-cell variability, even between members of the same strain under controlled homogeneous environments (Davidson & Surette, 2008). This individuality strengthens even more the notion of a highly personalized microbiome in each human or animal host, which ultimately affects host response to diet. Moreover, we often do not deal with the microbes per se; instead, we habitually deal with DNA nucleotide sequences obtained from these unique microbes at discrete time-points. In order to efficiently classify microbes based on molecular data (e.g., DNA sequences), scientists have developed categorization (or grouping) items and rules besides the idea of species, a term that is obviously ambiguous for organisms with eccentric reproductive strategies (Angert, 2005). The term Operational Taxonomic Unit (OTU) was invented in a context of numerical taxonomy (Sneath, 1957; Sokal & Sneath, 1963) and nowadays is mostly used to catalogue genetic sequences from marker genes (e.g., the 16S gene) based on nucleotide similarities. In short, similar sequences are catalogued within the same OTU, and therefore each OTU is thought to represent similar microorganisms. Alternatives to OTUs have been suggested (Callahan, McMurdie & Holmes, 2017) but the idea of using nucleotide similarities to catalogue microbes prevails. Note that the supposition that similar 16S sequences come from similar organisms is far from being true (Jaspers & Overmann, 2004).

Numerous studies have investigated the gut microbiota in relationship to diet, health and obesity. However, only few have looked at data from multiple studies to expose the impact of analysis strategies on this relationship. Lozupone et al. (2013) performed a meta-analysis of studies of the human microbiota (gut, oral, vaginal, skin and other) and showed that samples of the Western adult fecal microbiota clustered strongly by study. However, the authors did not discuss the potential contribution of diet in the differences in the microbiota and only used one approach to select OTUs. Walters, Xu & Knight (2014) performed a similar meta-analysis of human gut microbes associated with obesity and Inflammatory Bowel Disease and showed that specific microbial signatures of obesity were not consistent between studies. The authors suggested that this was due to differences in effect sizes and explained that some conditions such as inflammatory bowel disease are associated with more obvious differences in microbiota compared to other conditions such as obesity, whose association with the microbiota, accordingly to the authors, is less clear. The authors of this study also did not look at the contribution of diet to the observed differences and did not explore the use of different analysis strategies. A recent meta-analysis of samples from rodents and humans confirmed that diet can induce reproducible microbiome alterations but only focused on high-fat diets and did not assess the impact of analysis strategies on the relative abundance of taxa (Bisanz et al., 2019). Others have used a different approach looking only at the –published– abundance of a few selected bacterial groups (e.g. Bifidobacterium) and their relationship with dietary components (So et al., 2018; Wilson et al., 2019).

The mucin-degrader bacterium Akkermansia muciniphila is a good example to emphasize the relevance of these issues. This bacterium has anti-inflammatory and anti-obesity effects that are mediated by a close biochemical interaction with the host through the colonic mucus (Cani & De Vos, 2017). The relative abundance of this bacterium in feces ranges from 0.1% to up to 85%, and this variation is often considered to reflect a response to dietary components and health status (Garcia-Mazcorro et al., 2020). However, some of this variation may also be derived from the specific analysis strategy used during the analysis, although this issue has received little attention (Garcia-Mazcorro et al., 2020). The aim of this study was to explore how different analysis strategies impact the results from gut microbiological studies in a context of diet, health and obesity.

Materials & Methods

Ethical considerations

This study used 16S rRNA gene sequence data from public datasets published by our research group (Table 1), all of which received approval for the use of the animals.

Table 1 List of publications from which the data for this study came from†.

Publication	Number of samples	SRA Bioproject	Animal species	Sequencing Technique	Reported OTUs	
Peach (Noratto et al., 2014)	12	PRJNA217444	Zucker rats (obese)	Pyrosequencing	1,549 (approach not mentioned)	
Wheat (Garcia-Mazcorro et al., 2016)	30	PRJNA281761	db/db mice	MiSeq (Illumina)	8,686 (open)/1,302 (closed)	
Quinoa (Garcia-Mazcorro, Mills & Noratto, 2016)	9a	PRJNA299688	db/db mice	MiSeq	5,774 (open)	
Barley (Garcia-Mazcorro et al., 2017)	10a	PRJNA314690	db/db mice	MiSeq	5,366 (open)	
Cherry (Garcia-Mazcorro et al., 2018a)	44b	PRJNA407462	db/db mice (colon microbiota)	MiSeq	Not mentionedc	
Raspberry (Garcia-Mazcorro et al., 2018b)	27	PRJNA415476	db/db mice	MiSeq	675d(open)	
Apple (Garcia-Mazcorro et al., 2019)	32	PRJNA504388	Dawley Sprague rats	MiSeq	69,010 (1,339d) (open)	
Notes.

† In the original published articles, all of these studies used the 97% OTU files and a 97% similarity threshold. The compositional information about all diets is provided as Supplementary Information. All of these studies used the primers F515 (5′-GTGCCAGCMGCCGCGGTAA-3′) and R806 (5′-GGACTACHVGGGTWTCTAAT-3′) targeting the V4 region of the 16S rRNA gene, with the exception of the peach study that used the primers 28F (5′-GAGTTTGATCNTGGCTCAG-3′) and 519R (5′-GTNTTACNGCGGCKGCTG-3′) targeting the V1-V3 regions of the 16S rRNA gene. The Quick-DNA Fecal/Soil Microbe Miniprep Kit (Zymo Research) was used in all studies except for the peach study that used the QIAamp® PowerFecal® DNA kit (Qiagen).

a These studies (quinoa and barley) used the lean and obese controls from the paper published about whole-wheat by Garcia-Mazcorro et al. (2016).

b The associated BioProject does not contain all of these samples.

c The original cherry study did not report the number of OTUs. In this work, we report 2,439 to 138,203 OTUs for the cherry study, depending on the approach (see Table 2).

d After removal of very low abundant OTUs (i.e., OTUs with <0.005% of all reads). By default, the open approach in QIIME discards singletons (i.e., OTUs that appear only once).

16S gene sequencing data

We used 16S gene sequencing data from seven of our previous publications dealing with modulation of the gut microbiota using dietary interventions in animal obese models (Table 1). Performing a comparative analysis of data generated by the same research group is advantageous because technical variation is likely to be less compared to the variation obtained from multiple research groups. The sequencing procedure was performed at UC Davis for three studies, while the remaining four studies were sequenced at the Molecular Research LP. The Quick-DNA Fecal/Soil Microbe Miniprep Kit (D6010; Zymo Research) was used in all studies with only slight variations except for the peach study that used the QIAamp® PowerFecal® DNA kit (Qiagen). We did not use FastPrep (MP Biomedicals) for bead beating in any of these studies and this is important to clarify because some people consider the use of FastPrep essential for optimal lysis.

Six factors were studied across the samples: ‘dietary treatment’, with 11 levels; ‘study’, with seven levels, one for each study; ‘animal model’, with two levels: mice and rats; ‘sequencing technique’, with two levels: pyrosequencing and MiSeq; ‘obesity status’ at the time of sampling, with two levels: lean and obese; and ‘anatomical site’, with two levels: colon and feces. It is important to keep in mind that in the case that one factor is biologically significantly associated with a different microbiota in nature (e.g., obesity status), the existence of other interactive factors (e.g., dietary treatment) may mask the differences we observe during analysis. Unless otherwise stated, the data was analyzed with QIIME (Caporaso et al., 2010) v.1.8.0. After demultiplexing and quality filtering, we used the sequence file to assign 16S reads to OTUs based on the GreenGenes reference OTU sequence files (v.13.8, August 2013 release) clustered at 97% (99,322 sequences) and 99% similarity (203,452 sequences).

Assignment of 16S sequences to OTUs

Three strategies were used to assign the 16S sequences to OTUs. First, the conservative closed approach that discards sequences for not matching any sequence in the sequence reference data. Second, a de novo approach, which does not use a reference data set to cluster the sequences (Westcott & Schloss, 2015), thus being relatively free to depict the variety of sequences in a sample. Finally, an open approach that combines these two approaches (Rideout et al., 2014), first performing a closed approach followed by clustering of remaining sequences de novo. From this point on, we will refer to these strategies as the name of the clustering method (e.g., closed) and the reference file utilized (e.g., OTUs clustered at 97%). For example, ‘closed97’ and ‘closed99’ will refer to a closed approach using the 97% and the 99% OTU reference files, respectively. We only used the default 97% in the similarity option of the pick_otus.py script but changing this parameter can also drastically affect the results (more on this in “Similarity percentage between 16S rRNA gene sequences” in Supplemental Information).

Chimeras

Chimeras are possible combinations of two or more parent sequences that can inflate the observed OTUs and other diversity parameters, especially for de novo and open-reference strategies to select OTUs. However, there is a lack of consensus in chimera detection and removal and even the existence of true chimeras has been questioned. In this study, we subsampled the pynast alignment of representative sequences (one per OTU) separately for each study and applied the public version of uchime (v.4.2.40) available in Mothur for chimera detection. These analyses were performed on representative sequences from the de novo and open strategies. All the GreenGenes reference OTU sequence files (total: 14 files; OTUs clustered at different similarity percentages, from 99% similarity with 203,451 sequences, to 61% similarity with 22 sequences) were used to explore the behavior of uchime.

Taxonomic and diversity analyses

We combined all OTU tables (one for each of the seven studies) from the closed97 and the closed99 approaches into two separate OTU tables (one for each approach) and used these OTU tables for taxonomic classification and diversity analyses. The relative abundance of each taxon was calculated using all sequences (including possible chimeras and OTUs that appear only once) because it is impossible to detect true chimeras and the possible relevance of low abundant groups in the gut microbiome (Claussen et al., 2017). The unique fraction metric, or UniFrac, is a phylogenetic method for comparing microbial communities based on the phylogenetic distance between sets of taxa in a phylogenetic tree as the fraction of the branch length of the tree that leads to descendants from one environment or the other, but not both (Lozupone & Knight, 2005). Both weighted and unweighted UniFrac distances were used for comparing microbial communities because they can lead to different insights into factors that structure microbial communities (Lozupone et al., 2007). Principal Coordinate Analyses (PCoA) using these UniFrac distances were performed in QIIME and visualized using Emperor. Additionally, we used uniform manifold approximation and projection (UMAP, McInnes et al., 2018), a non-linear dimensionality reduction technique, to confirm the observed clusters, using the umap R package.

Prediction of functional profiling and phenotypes

PICRUSt (phylogenetic investigation of communities by reconstruction of unobserved states, Langille et al., 2013) was used to predict the functional profiling based on the 16S sequences. PICRUSt results were analyzed in STAMP (Parks & Beiko, 2010). Additionally, we used BugBase (Ward et al., 2017) to predict organism-level microbiome phenotypes for each study separately.

QIIME2

QIIME2 was introduced in 2017 on the basis of a plugin architecture that allows third parties to contribute functionality (Bolyen et al., 2019) and has been constantly updated ever since (more than 20 versions of QIIME2 have been released). This paper was conceived and started in QIIME1 in 2018 but because QIIME1 is no longer updated or supported, DADA2 (Callahan et al., 2016) and Deblur (Amir et al., 2017) were also used to select OTUs for a few selected studies in QIIME2, in part based on the observations made by Thompson et al. (2017) in a context of OTUs and sequence variants. One interesting feature of DADA2 is the removal of chimeric sequences.

Statistical analysis

The non-parametric analysis of similarities (ANOSIM) and the Adonis tests were used to determine whether the clustering of samples by a given factor (e.g., study) is statistically significant based on UniFrac distances, using the compare_categories.py script in QIIME with default number of permutations (999). In our experience, these tests usually have low sensitivity (they usually yield low p values even for weak clustering of samples), therefore it is informative to look at both the p values and the percentage of variation explained by the factor. We used the non-parametric Kruskal-Wallis to compare the number of OTUs between the different levels of any given factor, and also to compare the results from BugBase.

Results

We analyzed >15 million 16S reads from 164 samples from seven studies dealing with dietary interventions in obese laboratory animals (Table 1). In general, each study investigated the effect of different diets on the gut microbiota of obese laboratory animals and compared the results with data from control obese and/or lean animals (Table S3).

We detected 5,246 OTUs using the combined OTU table from the closed97 approach (n = 162), and 8,898 OTUs from the closed99 approach (n = 163) (two samples from the closed97 and one sample from the closed99 approach with the lowest number of sequences were discarded to generate a better assessment of diversity). This difference of ∼3,600 OTUs is substantial (70% more OTUs compared to the closed97 approach) and likely reflects the higher number of reference OTUs available for clustering the unknown 16S gene sequences. The number of detected OTUs from the closed97 approach was always lower, and with the exception of the peach study, the number of OTUs from the de novo approaches was always higher (Table 2). The percentage of possible chimeras in the OTU representative sequences ranged from 11% (wheat study) to 55% (apple study) for the de novo approach (median: 21%), and from 4% (quinoa study) to 45% (apple study) for the open approach (median: 13%). Interestingly, the percentage of possible chimeras reached a plateau in all the studies, where increasing the number of reference OTU sequences did no longer yield higher numbers of chimeras, and were lower for the open approach in all the studies with the exception of the raspberry study. No singletons (OTUs that appear only once) were found in any analysis using the open approach because the default script prevents it, but the de novo approach always showed ∼2 times higher percentage of singletons compared to the closed approach, except for the peach and the cherry study (Table 2). The factor ‘animal model’ (mice and rats) generated the highest number of significantly different OTUs, and ‘obesity status’ (obese and lean) the lowest (Table 3). The analysis of the combined OTU tables from the closed97 and the closed99 approach showed that only 161 and 165 OTUs (∼3% of all OTUs detected) were present in 50% of the samples, respectively. Very few OTUs (3-4) were present in >80% of samples using either approach, and no OTU was present in >90% of samples.

Table 2 Number of OTUs and relative proportions (i.e., percentages of 16S rRNA gene sequences) of the most abundant phyla.

Approach	OTUs	Singletons	Firmicutes	Bacteroidetes	Proteobacteria	Verrucomicrobia	Actinobacteria	Tenericutes	Cyanobacteria	
Peach study	
Closed97	758 ↓	114 (15%)	50.39%	40.84% ↑	6.09% ↑	2.37% ↑	0.072% ↑	0.128% ↓	0.065% ↓	
Closed99	1,074	192 (18%)	52.35% ↑	39.31%	5.77%	2.23%	0.069%	0.148%	0.077%	
De novo	1,549	147 (9%)	50.14% ↓	35.95%	4.28% ↓	1.67% ↓	0.042%	1.766%	0.112%	
Open97	1,603	NA	50.18%	35.15% ↓	4.29%	1.73%	0.040% ↓	1.766%	0.113% ↑	
Open99	1,680 ↑	NA	50.15%	39.95%	4.29%	1.73%	0.042%	1.775% ↑	0.112%	
SD:	0.88	2.45 ↑	0.85	0.31	0.02 ↓	0.85	0.02	
Wheat study	
Closed97	1,302 ↓	414 (32%)	45.81% ↓	32.52%	13.41% ↑	5.89% ↑	2.25% ↓	0.0249%	0.0282% ↓	
Closed99	2,008	734 (37%)	65.97% ↑	21.29% ↓	7.64%	2.06%	2.90%	0.0161% ↓	0.0456%	
De novo	37,474 ↑	28,466 (76%)	57.96%	31.82%	3.18% ↓	0.81% ↓	3.46%	0.3405%	0.0927%	
Open97	8,686	NA	58.73%	32.78%	3.22%	0.85%	3.53% ↑	0.3471%	0.0912%	
Open99	9,013	NA	58.73%	32.79% ↑	3.21%	0.85%	3.53% ↑	0.3476% ↑	0.0908%	
SD:	6.51 ↑	4.53	4.19	2.03	0.52	0.17	0.03 ↓	
Quinoa study	
Closed97	1,062 ↓	333 (31%)	55.34% ↓	30.29% ↑	4.68% ↑	7.20% ↑	2.20% ↓	0.051%	0.050%	
Closed99	1,606	578 (36%)	69.77% ↑	21.32% ↓	2.73%	3.29%	2.61%	0.045% ↓	0.070% ↑	
De novo	17,046 ↑	11,500 (67%)	63.70%	27.53%	2.45% ↓	1.04% ↓	3.30%	0.280% ↑	0.041%	
Open97	5,774	NA	64.08%	28.18%	2.46%	1.07%	3.35%	0.275%	0.037%	
Open99	5,976	NA	64.07%	28.17%	2.46%	1.07%	3.36% ↑	0.275%	0.035% ↓	
SD:	4.62 ↑	3.04	0.89	2.49	0.49	0.12	0.01 ↓	
Barley study	
Closed97	1,078 ↓	342 (32%)	53.50% ↓	32.38%	5.09% ↑	6.63% ↑	2.29% ↓	0.0157%	0.022% ↓	
Closed99	1,586	572 (36%)	62.54% ↑	29.04% ↓	3.01%	2.67%	2.60%	0.0131% ↓	0.053%	
De novo	15,599 ↑	10,429 (67%)	55.65%	36.03%	2.08%	1.20% ↓	3.25%	0.2859%	0.106% ↑	
Open97	5,366	NA	55.86%	36.75%	2.07%	1.23%	3.32%	0.2809%	0.104%	
Open99	5,594	NA	55.85%	36.77% ↑	2.06% ↓	1.23%	3.32% ↑	0.2812%	0.102%	
SD:	3.09	3.14 ↑	1.21	2.17	0.45	0.14	0.04 ↓	
Cherry study	
Closed97	2,439 ↓	1,942 (80%)	50.89%	36.60%	6.75% ↑	4.67%	0.180%	0.000% ↓	0.000% ↓	
Closed99	4,217	3,226 (77%)	51.68% ↑	36.00% ↓	6.18%	5.11% ↑	0.172% ↓	0.012%	0.001%	
De novo	138,203 ↑	64,000 (46%)	47.27% ↓	43.20% ↑	4.80%	3.46%	0.208%	0.079%	0.002%	
Open97	69,658	NA	48.53%	41.71%	4.73% ↓	2.90% ↓	0.208%	0.096% ↑	0.002% ↑	
Open99	70,886	NA	48.53%	41.75%	4.73% ↓	2.90% ↓	0.209% ↑	0.001%	0.002%	
SD:	1.68	3.12 ↑	0.90	0.98	0.02	0.05	0.00 ↓	
Raspberry study	
Closed97	2,751 ↓	722 (26%)	44.99% ↓	40.66% ↑	13.33% ↑	0.052% ↑	0.092%	0.002% ↓	0.001% ↓	
Closed99	4,433	1,231 (28%)	45.13%	40.53%	13.32%	0.052% ↑	0.092%	0.002% ↓	0.001% ↓	
De novo	92,486 ↑	69,306 (75%)	46.13% ↑	39.40% ↓	12.90% ↓	0.049% ↓	0.097% ↑	0.005% ↑	0.002% ↑	
Open97	21,243	NA	46.10%	39.73%	12.99%	0.049% ↓	0.088% ↓	0.004%	0.001% ↓	
Open99	21,834	NA	46.10%	39.72%	12.99%	0.049% ↓	0.088% ↓	0.004%	0.001% ↓	
SD:			0.55	0.56 ↑	0.19	0.002	0.004	0.001	0.0004 ↓	
Apple study	
Closed97	2,095 ↓	510 (24%)	55.32% ↓	39.17% ↑	3.48% ↓	0.499%	0.270%	0.002% ↓	0.542% ↑	
Closed99	3,363	907 (27%)	55.43%	38.74%	3.97% ↑	0.510% ↑	0.246% ↓	0.002% ↓	0.448%	
De novo	153,681 ↑	81,032 (53%)	61.48%	28.63% ↓	3.89%	0.279% ↓	0.432%	1.517%	0.361%	
Open97	69,010	NA	61.84% ↑	29.28%	3.96%	0.282%	0.435%	1.578% ↑	0.364%	
Open99	70,056	NA	61.83%	29.61%	3.96%	0.282%	0.436%	1.577%	0.364%	
SD:			3.25	5.14 ↑	0.19	0.12	0.09	0.79	0.07 ↓	
Notes.

The symbols (↑) and (↓) are used for the highest and the lowest value of each dataset, respectively. As explained in the main text, closed97 and closed99 refer to the closed OTU picking approach using the reference OTUs clustered at 97% (99,322 sequences) and 99% similarity (203,442 sequences), respectively. All these results were obtained using the default 97% in the similarity option of the pick_otus script (more on this in “Similarity percentage between 16S rRNA gene sequences” in Supplemental Information). The relative abundance of each phyla includes singletons and possible chimeras (see main text for more information about this). SD: standard deviation across all five OTU picking approaches.

Community membership structure

Table 2 shows an accurate numerical impression of relative abundances of all bacterial phyla across different OTU picking strategies, with up to 20% difference in relative abundance of some taxa, depending on the strategy analysis. However, Table 2 lacks a general view of how each taxon is represented across the different levels of each factor. A visual analysis of membership data revealed interesting patterns, for instance, the relative abundances of Firmicutes and Bacteroidetes were equally represented in each level of all six factors studied, while Cyanobacteria was poorly represented in samples from mice (Fig. 1). However, note that these results may also be misleading when considering the very few OTUs that were shared across all samples.

Table 3 Number of OTUs that were significantly differenta accordingly to the different factors.

Factor	FDR	Bonferroni	
Study (7 levels)	309	174	
Treatment (11 levels)	278	105	
Sequencing (two levels)	225	112	
Animal model (two levels)	134	53	
Anatomical site (two levels)	31	14	
Obesity (two levels)	4	3	
Notes.

a Defined as an adjusted p < 0.01 in non-parametric Kruskal-Wallis (adjusted for False Discovery Rate and Bonferroni).

Peach study

The peach study was the only one using 454 pyrosequencing, a different DNA extraction method, and a different primer set. As expected, an increase in quality threshold reduced the number of sequences that passed quality filtering. This is important because it has been suggested that a more stringent quality filtering helps to “reduce the number of spurious OTUs” (Buza et al., 2019), although in practice it is difficult to determine the exact quality threshold to differentiate “true” vs “false” OTUs (Edgar, 2017). We will not discuss this issue for Illumina platforms because defaults have already been established (Bokulich et al., 2013).

Figure 1 Proportions of relative abundances (percentages) of 16S reads.

The data is shown accordingly to the factors (A) animal model, (B) obesity status, (C) sequencing, (D) anatomical site, (E) study, and (F) treatment. Letters in bars represent different taxa (A: Firmicutes, B: Bacteroidetes, C: Proteobacteria, D: Verrucomicrobia, E: Deferribacteres, F: Actinobacteria, G: Cyanobacteria, and H: others). The bars do not represent the relative abundance of each taxon; instead, they show the proportion of 16S relative abundances from each level of the factors investigated. A balanced stacked bar denotes that the taxon was equally represented in each level of the factor (e.g., Firmicutes and Bacteroidetes between levels of all factors). An unbalanced stacked bar denotes that the taxon was more or less represented between the levels of the factor, for example Cyanobacteria between mice and rats, or between colon and feces.

Overall the different taxa remained similar in relative abundance but a higher base quality score (qual) threshold of 34 (default: qual 25) had a strong effect on the number of sequences available for OTU picking (62,971 sequences with qual 25 vs. 11,553 with qual 34) and consequently on the number of the OTUs discovered (758 OTUs with qual 25 vs 442 OTUs with qual 34 in the closed97 approach). The difference in qual threshold also had an effect in the proportions of Bacteroidetes that went down from ∼37% (qual 25) to 15% (qual 34), and Firmicutes that went up from ∼50% to 70%. This discrepancy was not related to the lower number of sequences available for OTU picking because a lower rarefaction in all other analyses (e.g., with qual 25) revealed similar relative proportions compared to the analysis with higher rarefaction depth. A higher base quality score threshold also affected the presence of some low abundant groups (e.g., at qual 34 Deferribacteres and Fusobacteria were not detected using all approaches).

In the peach study, Bacteroidetes displayed the highest standard deviation (SD) and showed the biggest difference (∼5%) in relative abundances, particularly between the open97 and the open99 approaches (Table 2). The difference between the lowest and the highest value was minimal for Firmicutes (∼2%), Proteobacteria (∼2%) and others (Table 2). Tenericutes showed the highest SD/average ratio (68.7), which implies that the variability was proportionally higher in this taxon, and Firmicutes the lowest (1.7), which implies that the variability was proportionally lower in this taxon. The detected phyla varied from 10 (closed97) to 13 (de novo97 and open97).

Wheat study

In the wheat study, Firmicutes displayed the highest SD and showed the biggest difference (∼20%) in relative abundances, particularly between the closed97 and the closed99 approaches (Table 2). This contrasts heavily with the biggest difference of ∼5% observed in the peach study. This difference in closed approaches was also noticeable in Bacteroidetes (∼11% difference), Proteobacteria (∼10% difference), and others (Table 2) and this was not related to rarefaction depth. This difference was not noticeable for the open and the de novo approaches where Firmicutes, Bacteroidetes, Proteobacteria and others showed very similar proportions in all cases (Table 2). Verrucomicrobia showed the highest SD/average ratio (107.9) and Firmicutes the lowest (11.3). The number of detected phyla varied between 10 (closed97) and 14 (de novo).

Quinoa study

Similar to the wheat study, in the quinoa study Firmicutes displayed the highest SD and showed the biggest difference (∼15%) in relative abundances, particularly in the closed approaches (Table 2). This difference in closed approaches was also noticeable in Bacteroidetes (∼11% difference) and minimal for Proteobacteria (∼2% difference) and others, and this was again not related to rarefaction depth. Also similar to the wheat study, this difference was not noticeable for the open and the de novo approaches where Firmicutes, Bacteroidetes, Proteobacteria and others showed very similar proportions in all cases (Table 2). Verrucomicrobia showed the highest SD/average ratio (101.8) and Firmicutes the lowest (7.3). The number of detected phyla varied from 8 (closed97) to 12 (de novo).

Barley study

In the barley study, Bacteroidetes displayed the highest SD but Firmicutes showed the biggest difference (∼9%) in relative abundances, particularly in the closed approach (Table 2). This difference in closed approaches was minimal in Bacteroidetes (∼3% difference), Proteobacteria (∼2% difference) and others and this was again not related to rarefaction depth. Also similar to the other studies, this difference was not noticeable in the de novo and open approaches (Table 2). Verrucomicrobia showed the highest SD/average ratio (91.9) and Firmicutes the lowest (5.5). The number of detected phyla varied from 8 (closed97) to 12 (de novo).

Cherry study

In contrast to the wheat, quinoa and barley studies, where the closed97 and the closed99 approaches showed different abundance of taxa, in the cherry study there was good agreement between the closed97 and the closed99 approaches for Firmicutes, Bacteroidetes, Proteobacteria and other groups, but these approaches showed dissimilar proportions compared to the open and the de novo approaches (Table 2). The cherry study was also interesting because the de novo approach showed the presence of bacterial groups (e.g., Nitrospirae, Chlorobi, Planctomycetes) that were not detectable using the closed and the open approaches, and this was likely not related to rarefaction depth because the open approach used similar thresholds. Bacteroidetes displayed the highest SD and showed the biggest difference (∼7%) in relative abundances. Tenericutes showed the highest SD/average ratio (100.9) and Firmicutes the lowest (3.4). The number of detected phyla varied from 14 (open99) to 22 (de novo). The use of DADA2 and Deblur in QIIME2 showed 1,329 and 1,263 OTUs, respectively. This is about half the lower number of OTUs (2,439 with closed97) obtained with the other approaches (Table 2).

Raspberry study

Unlike the other studies discussed above, the raspberry study was very interesting because it had the lowest variation among the different OTU picking approaches (0.56% difference in Bacteroidetes and 0.55% difference in Firmicutes between the lowest and the highest results), which was also reflected in the SD/average ratio (highest: 32.8 for Cyanobacteria, lowest: 1.2 for Firmicutes) (Table 2). Moreover, both the open97 and the open99 approaches detected as many as 41 different taxa at the phylum level, while the number of taxa in all other studies using the same approach only varied from 8 to 22. This strongly suggests that the microbiota data contained within each study is unique and may sometimes contain a high proportion of taxa that go unnoticed in other studies. This is very important in a context of the role of rare taxa in maintaining the stability of ecosystems (Jousset et al., 2017). DADA2 and Deblur showed 791 and 721 OTUs, respectively. This contrast with the numbers (2,751-92,486 OTUs) obtained from all approaches (Table 2).

Apple study

In the apple study, Bacteroidetes showed the highest variation and also the biggest difference (∼11%) especially between the closed and the de novo approaches (Table 2). This difference between the two approaches was also noticeable in Firmicutes and Tenericutes (Table 2). Tenericutes showed the highest SD/average ratio (77.3) and Proteobacteria the lowest (4.9). The number of detected phyla varied from 15 to 35, thus making the apple study the second study more variable after the raspberry study. The use of DADA2 and Deblur in QIIME2 showed 1,357 and 1,149 OTUs, respectively. This contrast with the numbers (2,095-153,681 OTUs) obtained from all approaches (Table 2).

Firmicutes/Bacteroidetes ratio

The first paper that discussed about the possible usefulness of the Firmicutes/Bacteroidetes ratio was published by Ley et al. (2005), where the showed that obese mice had 50% more Firmicutes, with a proportionally lower abundance of Bacteroidetes, thus leading to a higher Firmicutes/Bacteroidetes ratio compared to lean mice. However, each phylum is composed by hundreds of different species, and therefore the ratio between the two has little significance as discussed elsewhere (Delzenne & Cani, 2011). In this current study, this ratio varied from 1.1 to 3.3 due to the differences in relative abundance in the two phyla (Table 2).

The phylum Verrucomicrobia

Firmicutes and Bacteroidetes are usually the most abundant phyla in the gut microbiota and they have attracted most of the attention. However, other taxa deserve attention to better comprehend the functioning of the gut microbial ecosystem. A. muciniphila is a taxon that has generated interest as a new generation probiotic candidate to help obese patients and is considered to be a member of Verrucomicrobia based on 16S gene analysis. Akkermansia was detected in ∼15% the samples (25/164) with an average of 3% and it was more represented in colon samples (Fig. 1). Interestingly, the whole Verrucomicrobia phylum also had an average of 3%, which implies that most or all Verrucomicrobia was represented by Akkermansia. If we translate the proportions of 16S reads into numbers, a 3% would represent approximately 300 million cells (3 ×108) in a hypothetical environment of 1 ×1010 cells/g of intestinal contents. While this conversion is not necessarily accurate, the high numbers may bear some ecological relevance if one considers that as low as 1,000 cells of other microbes are enough to thrive (Nilsson, Kari & Steele-Mortimer, 2019). In three studies, the abundance of Verrucomicrobia was >5 times higher using a closed approach compared to the other approaches (Table 2).

The phylum Cyanobacteria

Another group of interest for gut health is Cyanobacteria (Barcena et al., 2019), which is often considered a contaminant and removed from 16S gene analysis. However, Ley et al. (2005) showed a deep-branching clade of Cyanobacteria in the guts of animals and mentioned that they may represent descendants of non-photosynthetic ancestral Cyanobacteria that have become adapted to life in the mammalian gut. Here, Cyanobacteria was detected in ∼50% the samples (74/164) and was always low in relative abundance (<0.5%) (Table 2). While 0.5% may be considered low, this percentage implies approximately 50 million (5 ×107) cells in the same hypothetical environment of 1 ×1010 cells/g of intestinal contents as discussed above.

Analyses of UniFrac distances

UniFrac analyses from the closed97 approach

The first two coordinates explained 18% of the variation using unweighted distances (Fig. 2), which is similar to other studies that have shown that the first two coordinates explained 23% of the variation in a data set of unweighted UniFrac distances from samples of the human microbiota (Lozupone et al., 2013). Accordingly to ANOSIM tests on unweighted distances, study, sequencing procedure, animal model, and dietary treatment were the factors most highly associated with the differences in the bacterial communities (Fig. 2, Table 4). UMAP confirmed the clustering by animal model and study (Supplemental Information). This is interesting because others have also shown a strong study effect (Lozupone et al., 2013).

Figure 2 PCoA plots of unweighted UniFrac distances using data from the closed approach using the reference OTUs sequence file at 97% similarity (closed97 approach).

Each point represents a sample from one of the studies detailed in Table 1, and the plots highlight the effect of (A) animal model, (B) obesity status, (C) sequencing technique, (D) anatomical site, (E) study and (F) treatment. The labels “lean” and “obese” refer to lean and obese controls. The ten most abundant bacterial groups are superimposed in all plots (the bigger the circle, the bigger the relative abundance of each taxa) and labelled with numbers on the first plot (1: Ruminococcaceae, 2: Bacteroides, 3: Clostridiales, 4: Lachnospiraceae, 5: Oscillospira, 6: Enterobacteriaceae, 7: S24-7, 8: Lactobacillus, 9: Akkermansia, 10: Allobaculum) to show that clustering of samples is driven by specific bacterial groups that have previously been shown to influence (or be influenced by) health status, such as Akkermansia. The values for each axis are only shown in A to facilitate viewing. These plots were built using a rarefaction depth of 100 sequences per sample to account for as many samples as possible (only two samples were left out using this rarefaction depth). The UniFrac data was obtained from the closed97 approach but other approaches may reveal other patterns (see Tables 4 and 5).

Table 4 Summary of results for all samples (n = 162) from the Adonis and ANOSIM tests for comparing categories using UniFrac data from the closed97 approach.

	Adonis	ANOSIM	
	Unweighted	Weighted	Unweighted	Weighted	
Study	P < 0.001	P < 0.001	P = 0.001	P = 0.001	
	R2 = 24.8%	R2 = 19.8%	R = 66.3	R = 31.5	
Model	P < 0.001	P < 0.001	P = 0.001	P = 0.145	
	R2 = 8.8%	R2 = 7.4%	R = 56.7	R = 4.5	
Sequencing	P < 0.001	P < 0.01	P = 0.001	P = 0.612	
	R2 = 5.5%	R2 = 3.2%	R = 57.8	R =  − 3.2	
Treatment	P < 0.001	P < 0.001	P = 0.001	P = 0.001	
	R2 = 20.4%	R2 = 21.1%	R = 34.2	R = 20.1	
Obesity	P < 0.001	P < 0.001	P = 0.023	P = 0.001	
	R2 = 1.9%	R2 = 4.4%	R = 9.5	R = 23.6	
Site	P < 0.001	P < 0.01	P = 0.702	P = 0.868	
	R2 = 3.8%	R2 = 2.2%	R =  − 1.9	R =  − 4.7	
Notes.

According to QIIME documentation, the R2 value (effect size) calculated with Adonis test shows the percentage of variation explained by the factor (e.g., the factor ‘study’ explained 24.8% of the variation in the unweighted distances), and the R statistic calculated with ANOSIM test reflects the degree of dissimilarity (an R value near 1, or 100, means that there is dissimilarity between the groups, while an R value near 0 indicates no significant dissimilarity between the groups). A rarefaction depth of 100 sequences per sample to account for as many samples as possible (only two samples were left out using this rarefaction depth). A total of 999 permutations were used to calculate the statistics.

In this study, the first 2 coordinates explained 34% of the variation using the weighted UniFrac distances, which is about twice the variation explained by the first two axes using unweighted UniFrac, but produced a very similar clustering of samples. Accordingly to ANOSIM tests, the effect of study, animal model, and sequencing procedure explained much smaller proportion of the variability in the data using weighted distances, while obesity status explained a higher proportion of the variation (23.6% vs 9.5% using unweighted distances, Table 4).

UniFrac analyses from the closed99 approach

The analysis of unweighted UniFrac distances from the closed99 approach revealed similar results compared to the closed97 regarding the relative contribution of each factor in explaining the variability in the data (Fig. 3 and Table 5). The first two coordinates explained 25% of the variation (Fig. 3), which is even closer to the results of Lozupone et al. (2013). Interestingly, Akkermansia was not part of the ten most abundant groups (Fig. 3) and the factor ‘study’ explained most of the dissimilarity in the communities using the unweighted UniFrac data accordingly to the ANOSIM test (Table 5). In our experience it is uncommon that one single factor explains that much of the variability between microbial communities. Similar to the analysis using the closed97 approach, the clustering of samples was very similar but the effect of ‘study’, ‘animal model’ and ‘sequencing procedure’ at explaining the variability in the data was lower using weighted UniFrac, while obesity status explained much more of the variability in the data (Table 5).

Figure 3 PCoA plots of unweighted UniFrac distances using data from the closed approach using the reference OTUs sequence file at 99% similarity (closed99 approach).

Each point represents a sample from one of the studies detailed in Table 1, and the plots highlight the effect of (A) animal model, (B) obesity status, (C) sequencing technique, (D) anatomical site, (E) study and (F) treatment. The labels “lean” and “obese” refer to lean and obese controls. The ten most abundant bacterial groups are superimposed in all plots (the bigger the circle, the bigger the relative abundance of each taxa) and labelled with numbers on the first plot (1: Ruminococcaceae, 2: Bacteroides, 3: Clostridiales, 4: Lachnospiraceae, 5: Oscillospira, 6: Enterobacteriaceae, 7: S24-7, 8: Lactobacillus, 9: Parabacteroides, 10: Allobaculum) to show that clustering of samples is driven by specific bacterial groups that have previously been shown to influence (or be influenced by) health status, such as Ruminococcaceae. The values for each axis are only shown in A to facilitate viewing. These plots were built using a rarefaction depth of 440 sequences per sample to account for as many samples as possible (only two samples were left out using this rarefaction depth). The UniFrac data was obtained from the closed99 approach but other approaches may reveal other patterns (see Tables 4 and 5).

Table 5 Summary of results for all samples (n = 163) from the Adonis and ANOSIM tests for comparing categories using UniFrac data from the closed99 approach.

	Adonis	ANOSIM	
	Unweighted	Weighted	Unweighted	Weighted	
Study	P < 0.001	P < 0.001	P = 0.001	P = 0.001	
	R2 = 33.2%	R2 = 33.4%	R = 80.1	R = 43.5	
Model	P < 0.01	P < 0.001	P = 0.001	P = 0.016	
	R2 = 11.9%	R2 = 10.4%	R = 53.9	R = 9.8	
Sequencing	P < 0.001	P < 0.001	P = 0.001	P = 0.529	
	R2 = 6.7%	R2 = 3.7%	R = 61.6	R =  − 2	
Treatment	P < 0.001	P < 0.001	P = 0.001	P = 0.001	
	R2 = 24.6%	R2 = 28.2%	R = 33.8	R = 25	
Obesity	P < 0.001	P < 0.001	P = 0.237	P = 0.001	
	R2 = 1.8%	R2 = 5.5%	R = 3.1	R = 20.6	
Site	P < 0.001	P < 0.01	P = 0.960	P = 0.953	
	R2 = 5.1%	R2 = 2.4%	R =  − 5.3	R =  − 6.3	
Notes.

According to QIIME documentation, the R2 value (effect size) calculated with Adonis test shows the percentage of variation explained by the factor (e.g., the factor ‘study’ explained 33.2% of the variation in the unweighted distances), and the R statistic calculated with ANOSIM test reflects the degree of dissimilarity (an R value near 1, or 100, means that there is dissimilarity between the groups, while an R value near 0 indicates no significant dissimilarity between the groups). A rarefaction depth of 440 sequences per sample to account for as many samples as possible (only one sample was left out using this rarefaction depth). A total of 999 permutations were used to calculate the statistics.

UniFrac analyses from closed97 approach on mice samples

To discover any additional pattern or association between the microbial communities, we performed a separate analysis of mice samples only (n = 120). Briefly, unweighted UniFrac analyses using closed97 showed that ‘treatment’ and ‘study’ were the most important factors, each explaining almost half of the dissimilarity in the data accordingly to ANOSIM tests, while the use of weighted UniFrac distances revealed a lower contribution of ‘study’ and ‘treatment’ and a higher contribution of obesity status at differentiating microbial communities (more about this in “UniFrac analyses from closed97 approach on mice samples” in Supplemental Information).

Prediction of functional profiling

PICRUSt revealed a total of 217 features that were significantly different between different studies, 211 between different treatments, 145 between sequencing techniques, 136 between mice and rats, 69 between lean and obese, and 32 between feces and colon contents (adjusted P < 0.05, see ‘PICRUSt results’ in Supplemental Information). This is interesting because the factor ‘study’ was also associated with the highest amount of variability. The weighted Nearest Sequenced Taxon Index (weighted NSTI) score was developed to summarize the extent to which microorganisms in a given sample are related to sequenced genomes (e.g., a NSTI score of 0.03 indicates that the genomes of the microbes were well represented and that the average microbe in the sample can be predicted using a relative from the same “species”). Higher NSTI values indicate a lower accuracy of the predictions. Here, NSTI scores varied widely, from a minimum of 0.03 (sample W11, feces from lean mice sequenced with MiSeq, wheat study) to a maximum of 0.34 (sample W32, feces from obese mice on quinoa sequences with MiSeq, quinoa study). There was no indication to suggest that any group of sequences were associated with lower or higher NSTI scores.

Prediction of organism-level microbiome phenotypes

BugBase revealed interesting differences in phenotypes within each study. For example, obese rats in a high-fat diet had higher proportions of bacteria with the potential of forming biofilms among the different treatments in the apple study, lean mice had higher proportions of bacteria containing mobile elements in the barley study, and obese mice had higher proportions of stress-tolerant bacteria in the cherry study (more on this in ‘BugBase results’ in Supplemental Information). Overall, these results also indicate that each study showed unique peculiarities, ultimately derived from the molecular composition of the 16S gene sequences at any given time-point within each particular study.

Discussion

Bacteria and other microorganisms are artificially classified for various reasons, for example to understand their relationship with other organisms. In most scientific publications, it is implicitly assumed that this artificial classification yields biologically relevant groups that exist in nature at real, normal abundances in a state of “normobiosis”. This cannot be better exemplified by the widespread use of the opposite term “dysbiosis”, a term that is not only inaccurate but also misleading (Brussow, 2020). However, the true abundance of any bacterial group in nature is very difficult to determine with accuracy, due to both biological facts (Davidson & Surette, 2008; Jaspers & Overmann, 2004), methodological biases, and the many remaining unknowns in microbiome research (Thomas & Segata, 2019). This work provides clear evidence of this issue by showing that different analysis strategies generate up to 20% difference in relative abundance of some taxa from the same samples, depending on the strategy.

Our work also sheds light into the issue of the number of bacterial species in the gut. Considering the lowest sequence abundance threshold of one, it has been suggested that our planet harbor about 10,000 bacterial species (Caporaso et al., 2011) but others have suggested 5.6 million OTUs as the lower bound of the microbial diversity on Earth (Rideout et al., 2014). On the other hand, the human microbiome is thought to harbor 15,000 to 36,000 species (Frank et al., 2007) yet others have suggested only 4,930 species of bacteria in that environment (Pasolli et al., 2019). In technical terms, the number of OTUs is dependent on the number of sequences that pass quality filtering (i.e., the higher the quality threshold, the lower the number of sequences available to catalogue). However, the number of OTUs is also dependent on the specific strategy used to catalogue sequences into OTUs. For instance, in this study the number of OTUs from the closed97 approach was always lower, and with one exception, the number of OTUs from the de novo approaches was always higher. While the higher number of OTUs detected by the de novo approach may not be biologically relevant (Edgar, 2017), this may deserve more thought if we consider the unexplored variety of microbes in nature.

As mentioned above, bacteria and other microbes display high levels of cell-to-cell variability; in other words, these organisms show high levels of individuality, which implies that every host carries a unique set of microbes. In support of this hypothesis, the analysis of the combined OTU tables from the closed97 and the closed99 approach showed that only very few OTUs were shared among the samples (3–4 OTUs were present in >80% of samples using either approach, and no OTU was present in >90% of samples). These results strongly suggest a unique microbial profile (i.e., variety of 16S sequences) in each sample. More importantly, the fact that each bacterial cell displays a high level of individuality raises doubts about the relevance of nucleotide similarities in the 16S gene (or even whole genomes, see Lukjancenko, Wassenaar & Ussery, 2010) to estimate or predict similarities in nature. In other words, it is feasible and very likely that even those same OTUs that were apparently shared by multiple samples represent in fact different microorganisms.

The analysis of 16S data from each study revealed interesting patterns. However, there were two studies that attracted our attention. The raspberry study was very interesting because it had the lowest variation in relative abundance of taxa among the different OTU picking approaches, which was also reflected in the SD/average ratio. Also, in the raspberry study both the open97 and the open99 approaches detected as many as 41 different taxa at the phylum level, while the number of taxa in all other studies using the same approach only varied from 8 to 22. This strongly suggests that the microbiota data contained within each study is unique and may sometimes contain a high proportion of taxa that go unnoticed in other studies. This is very important in a context of the role of rare taxa in maintaining the stability of ecosystems (Jousset et al., 2017).

This paper shows interesting results about the impact of the analysis strategies in the relative abundance of Verrucomicrobia and Cyanobacteria and these taxa are also important in a context of low abundant or rare taxa (or OTUs in datasets). Several strategies have been used to remove the so-called rare sequences. Needham, Sachdeva & Fuhrman (2017) used 99% de novo OTUs (i.e., OTUs generated using 99% similarity without the use of a reference database) that exceeded a threshold of 0.4% in relative abundance without providing any rationale for using this threshold. In contrast, Navas-Molina et al. (2013) filtered the data based on the proportions of sequences represented by the OTUs (the authors suggested to remove OTUs that represents less than 0.005% of all sequence reads based on the analyses by Bokulich et al., 2013) and Lozupone et al. (2013) discarded whole samples that did not have at least 100 sequences after quality filtering and OTU assignment. Although the decision of removing rare sequences is often based on rational arguments (Bokulich et al., 2013) it is still arbitrary to choose one strategy or another. This is important because the so-called rare microbes may in fact be keystone species that can regulate entire microbial environments, including host-associated microbiomes (Jousset et al., 2017).

The factor(s) that better explain the differences in gut microbial communities are of interest for various disciplines in biomedical sciences. For instance, the results of this current work showed that the factor ‘animal model’ generated the highest number of significantly different OTUs, while ‘obesity status’ the lowest. Unlike other studies suggesting that microbial signatures of obesity are not consistent between studies because of effect sizes (Walters, Xu & Knight, 2014), in this study the lack of statistical difference in gut microbiota between obese and lean is most likely due to the fact that the obese rodents were subject to different dietary treatments, thus making difficult to dissect between obesity status and dietary treatment. In this regard, the results of the analysis of UniFrac distances are particularly important considering the thoughts discussed by Lozupone et al. (2007) that explained that unweighted UniFrac is better suited to detect effects of different “founding” populations (e.g., the source of bacteria that first colonize the gut of newborn individuals and factors that are restrictive for microbial growth such as temperature). In contrast, they suggested that weighted UniFrac can more accurately reveal the effects of more transient factors, for example nutrient availability. Host genetics can be considered as a founding effect that since the moment of birth imposes certain restrictions to growth of some microbes, and in our analyses the factor ‘animal model’ explained high proportion of the variation when using unweighted distances. In contrast, the analysis of weighted distances revealed that the effect of obesity status at sampling explained much more variability in the data compared to unweighted analyses, a result that also makes sense if we consider obesity as a more transient condition that suddenly appeared in the host-microbiota relationship. More research is needed to better investigate the relationship of the different factors as well as their interactions in the gut microbiota.

This paper is not free of limitations. For instance, the transport, storage and DNA extraction methods can drastically affect the reported microbial community structure (Martinez et al., 2019). These and other factors should be considered when studying the microbial composition of gut samples. Also, there is often overlap between factors affecting our view of the gut microbiota, for instance in this study the weight of the factor “sequencing technique” may be masked by the use of different primers or DNA extraction method, and obesity status influenced the dietary regimen (i.e., obese animals are more often subject to dietary manipulation). More importantly, the most recent ways to catalogue marker 16S genes (e.g., DADA2) are offering more accurate but also more conservative views of microbial diversity that may need radical conceptual improvements, for example considering the evolution of the non-coding strand of the 16S rRNA gene (Garcia-Mazcorro & Barcenas-Walls, 2016).

Conclusions

In summary, diet and the gut microbiota are strongly related to host health and their interactions are “convoluted and multi-faceted” (Xu & Knight, 2015). This paper aimed to contribute to current knowledge about the impact of using different analysis strategies in the relationship between the gut microbiota, diet and obesity. The results show that the use of different strategies to select OTUs have an impact on the relative proportions of bacterial taxa, particularly when using a closed OTU picking approach. This may be due to the higher number of reference OTUs available for clustering the unknown 16S gene sequences as well as the number of used reads between closed97 and closed99 strategies. We also demonstrated the impact of using OTU reference sequences clustered at different similarity percent and confirmed previous observations regarding a strong study effect. We invite others to consider and further expand the feasible possibility that the variations among studies are related to the individuality of bacteria. Overall, the results are useful to guide future research and meta-analyses aiming to investigate the complex relationship between diet, health, and the gut microbiota.

Supplemental Information

Supplemental Information 1 Supplemental Information, Figures, and Tables

Click here for additional data file.

Supplemental Information 2 OTU table closed97

Click here for additional data file.

Supplemental Information 3 OTU table closed99

Click here for additional data file.

We thank all developers and users of QIIME, PICRUSt, STAMP and BugBase, and especially Jose Pablo Gomez from the Center for Animal Disease Modeling and Surveillance at UC Davis for his valuable help with UMAP.

Additional Information and Declarations

Competing Interests

Author Contributions

Data Availability

Jose F. Garcia-Mazcorro is an employee of MNA de Mexico, a company of Animal Nutrition.

Jose F. Garcia-Mazcorro conceived and designed the experiments, performed the experiments, analyzed the data, prepared figures and/or tables, authored or reviewed drafts of the paper, and approved the final draft.

Jorge R. Kawas and Susanne Mertens-Talcott conceived and designed the experiments, prepared figures and/or tables, authored or reviewed drafts of the paper, and approved the final draft.

Cuauhtemoc Licona Cassani and Giuliana Noratto performed the experiments, analyzed the data, authored or reviewed drafts of the paper, and approved the final draft.

The following information was supplied regarding data availability:

Data used in this publication are available at SRA NCBI: PRJNA217444, PRJNA281761, PRJNA299688, PRJNA314690, PRJNA407462, PRJNA415476, PRJNA504388.

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
