# Peer review of "Different analysis strategies of 16S rRNA gene data from rodent studies generate contrasting views of gut bacterial communities associated with diet, health and obesity"

_PeerJ, doi:10.7717/peerj.10372_

## Round 0.1 · original submission · Major Revisions

All reviewers have been positive about your study, but they've suggested a number of additional analyses, or, rather, modifications of existing analyses, and I agree that these might make the paper even better.

·

Basic reporting

The article is written in rather clear and technically correct language. The introduction and background revealed the problem of reproducibility in microbiome studies and main factors that may have impact on the result of such investigations. However, please try to keep the same format of in-text citations, preferably with comma before year (please compare 51-52 and 72 lines). Also I believe that not several but numerous studies have investigated the relationship of microbiome contents and diet, health and obesity (line 79) especially if we would search not only for studies done in humans but also in rodents.

I suggest to improve some phrases to ensure that an international audience can clearly understand the text. Some examples where the language could be improved include lines 23, 50 – the current phrasing makes comprehension difficult.

The shown results are sufficient to support the conclusions made by the authors.

The links to the raw data are supplied in acceptable form.

Experimental design

The research question is well-defined, relevant and very important. I suggest adding to the description of the studies which region of the 16S rRNA gene was sequenced and with which primers set.

Also used DNA extraction technique may be very relevant factor to study across the samples too. (Have you tested this factor? It could be added to the supplementary materials.) As it is known that method of DNA extraction may affect the reported microbial community structure it may be interesting to test this hypothesis.

Can you please, provide explanation why in some cases you have used QIIME which is not supported for several years instead of QIIME2?

Also you have tested influence of the factor ‘sequencing technique’, with two levels: pyrosequencing and MiSeq across the samples. As you wrote that there was only one study utilizing pyrosequencing and a set of primers that differs from all other studies it is worth to emphasize that the effect of the factor ‘sequencing technique’ can be observed for several reasons (different sequencing technique, different set of primers, the study, or the combination of several of them).

Moreover, I suggest to test whether the quantitative levels of microbial taxa differed by sample source with more approaches then only PCoA. Especially, pay attention to the UMAP (Uniform Manifold Approximation and Projection) algorithm for dimensional reduction of diversity within large amounts of data by non-linear multidimensional clustering (McInnes et al., 2018) which is appropriate for such type of data and started widely used for such type of analysis.

Furthermore, on the PCoA are two the most definable clusters that are species-specific (mice and rats), it would be interesting to inspect only the mouse samples in order to find some more relevant clustering factors as they can be masked with the impact of the species specificity. It could be added to the supplements.

References:
McInnes L, Healy J, Saul N, Grossberger L. 2018. UMAP: Uniform Manifold Approximation and Projection. Journal of Open Source Software. 3:861. DOI:10.21105/joss.00861.

Validity of the findings

All the underlying data have been provided but I suggest to use not only PCoA but also UMAP for the patterns’ search.The aim of the study to assess the impact of different analysis strategies and several other factors on the resulting microbiome composition is achieved. The conclusions are well stated, and they are connected with the investigated question.

Additional comments

1. Please pay attention that it is commonly suggested to test whether the quantitative levels of microbial taxa differed by sample source with multiple approaches including UMAP.
2. Please consider correcting the in-text citations and some hard for the comprehension phrases.
3. Add a little bit more relevant data about the studies (which region of the 16S rRNA gene was sequenced and with which primers set; the DNA extraction method in order to have an ability to test if there is batch-effect due to this factor).
4. It would be valuable to look for influence of the factor of DNA extraction method on the microbial community structure.
5. Please, provide in the text explanation why in some cases you have used QIIME which is not supported for several years instead of QIIME2.
6.I suggest to inspect additionally only the mouse samples in order to search for some more relevant clustering factors as they can be masked with the impact of the species specificity.

Reviewer 2 ·

Basic reporting

The manuscript is clearly written and good structured. Raw data is available.

Experimental design

Research question is well defined, all analysis methods are described in sufficient detail.

Validity of the findings

This is very interesting and useful study about the influence of analysis parameters on the results. I do not have any major questions, only some minor:
1) For close-reference OTU-picking strategies does it have sense to add the number (or percent) of unused reads in each analysis? Could it appears, that only the minor fraction of reads was considered by these approaches? Also could the difference of the number of used reads between Closed97 and Closed99 strategies produce such changes in predicted bacteria phyla proportions shown in Table2? Because De novo and Open-reference strategies that use all reads give much more consistent phyla proportions.
2) It could be advantageous to add figures with PCoA using weighted UniFrac, because authors mentioned that it gives results that differs from obtained with unweighted one.
3) The number of observed OTU can be highly inflated because of chimeras, did authors consider to use any chimera filtering methods, especially for De novo and open-reference strategies?
One additional moment, that potentially is out of the planned scope of the manuscript. It could be very interesting to extend the analyses of the “study effect” to consider not only close-reference strategies, but also open-reference and especially DADA2 (of course for studies that use same 16s rRNA region).

·

Basic reporting

The manuscript is written in a good English with the sufficient amount of background information. It has the professional structure and self-contained.

Experimental design

The research question is well defined, although it is very broad and requires much more data to be analyzed to be answered in full. The title is somewhat misleading as only rodent gut microbiomes were studied and it is not clear what health conditions other than obesity were implied. The current manuscript is an exploratory analysis of effects of various clustering strategies on OTU assignment to 16S sequences and factors relevant for explanation of variations in the data.

Validity of the findings

Authors used several strategies on binning 16S sequences to OTU with QIIME1 software and GreenGenes reference database and showed that the results are highly unstable. This is consistent with other reports (like Prodan et al. 2020 PLOS ONE, Alimeida et al. 2018 Gigascience) and general knowledge in microbiome community. Although the authors provided the links to 16S reads in the SRA it would be highly beneficial to attach processed OTU abundance tables as the direct foundation of the conclusions.

The factor analysis is consistent with the current knowledge, but with only 7 studies it is limited for generalization. Especially for factors “sequencing technique” and “anatomical site” because of the strong confounding effect with the "study" factor.

Additional comments

1) Although 16S sequencing data is provided through NCBI SRA it would be good to add full OTU tables in the supplement.
2) I suggest changing title to better represent narrow scope of the research as only rodent gut microbiome was studied and the only health condition was obesity.
3) On figures among treatments levels indicated “obese” and “lean”. Are they control groups treatments? If this is true – are control obese and lean mice had different treatment? In the table S1 indicated several diets for control groups, but in all studies lean and obese mice had the same diet. Please check your factor levels assignment.
4) To put more weight for the factor analysis I suggest trying it only on control groups as it should show the study effect clearer.
5) On line 186 indicated n=164 samples in the study although on line 191 used n=162 and n=163 samples. Why 1-2 samples were omitted?
6) The line 346 stated that Cyanobacteria “showed the lowest SD among the OTU picking strategies”, however it has very low abundance and therefore low SD. It seems that this phrase could be deleted as do not lead to any conclusion.
7) The line 360 states that explained variation could be found in the Table 4. However it does not contain any information on the explained variation.
8) Line 447 “More importantly, the fact that each bacterial cell displays a high level of individuality raises doubts about the relevance of nucleotide similarities to estimate or predict similarities in nature.” It is not clear what does it mean exactly. Does it mean that bacteria with identical 16S can have substantial variance in genomes or bacteria with identical genome could exhibit different phenotype? Although it is a discussion it should be stated clearer.
9) In the Table 2 phrase “This data was obtained from the analysis of sequences using a 97% similarity against the reference OTU files.” is unclear as this table includes data from different approaches.
10) In Table 4 it is not clear what percents are shown.
11) Line 421 “Brüssow 2019” should be 2020.

---

## Round 0.2 · Minor Revisions

There are several minor remaining editorial comments.

·

Basic reporting

The article is written in clear, unambiguous, and technically correct language. The introduction and background revealed the problem of reproducibility in microbiome studies and main factors that may have impact on the result of such investigations.
The shown results are sufficient to support the conclusions made by the authors.
The supplied raw data are sufficient and are referenced in acceptable form. Figures are relevant, well labelled and described.

Experimental design

The methods are described with sufficient detail and information to replicate. And they are enough for the conclusions made.
Unfortunately authors ignored the suggestion to use more methods to search for clusters besides PCoA to see whether received clusters are stable (I’ve suggested to use UMAP (McInnes et al., 2018). As well as the suggestion to inspect only mouse samples on the PCoA in order to find more clusters masked by strong impact of the species specificity.

McInnes L, Healy J, Saul N, Grossberger L. 2018. UMAP: Uniform Manifold Approximation and Projection. Journal of Open Source Software. 3:861. DOI:10.21105/joss.00861.

Validity of the findings

All the underlying data have been provided but I suggest to use not only PCoA but also UMAP for the patterns’ search to confirm the stability of the result.The aim of the study to assess the impact of different analysis strategies and several other factors on the resulting microbiome composition is achieved. The conclusions are well stated, and they are connected with the investigated question.

Additional comments

Most of the comments were taken into account (the ignored suggestions were named above) and the questions raised were answered.

Reviewer 2 ·

Basic reporting

no comment

Experimental design

no comment

Validity of the findings

no comment

Additional comments

Authors gave answers to all comments and i suppose that the manuscript can be published now.

·

Basic reporting

Authors improved parts of manuscript that required clarifications. The language is clear and professional. Article is structured in appropriate way for publication. The necessary references and supplementary raw data are included. The manuscript is ready for publishing although I have one small suggestion (see General comments).

Experimental design

no comment

Validity of the findings

no comment

Additional comments

On Figure 2F among treatments there are “Lean” and “Obese”. It looks confusing. Although it could be assumed that these categories represent lean and obese mice in control groups I suggest to add explicit explanation to the description of the figure.

---

## Round 0.3 · accepted · Accept

I think the remaning concerns ave been resolved.